# Quality of Life and Well-Being in Women with Tetany Syndrome in the Context of Anxiousness and Stress Vulnerability

**DOI:** 10.3390/brainsci15040358

**Published:** 2025-03-30

**Authors:** Marta Górna, Zuzana Rojková

**Affiliations:** Faculty of Arts, University of Ss. Cyril and Methodius in Trnava, 91701 Trnava, Slovakia; zuzana.rojkova@ucm.sk

**Keywords:** tetany syndrome, quality of life, well-being, anxiousness, vulnerability to stress

## Abstract

**Background/Objectives:** This paper deals with quality of life (QoL), mental well-being (WB), anxiousness, and stress vulnerability in women with tetany syndrome (TS) in comparison with the population without the syndrome. The aim is to investigate the individual or combined effects of anxiousness, stress vulnerability, and tetany syndrome diagnosis on quality of life and well-being in women. **Methods**: The research sample was composed of 144 female (in terms of sex) respondents with a diagnosis of tetany syndrome and 123 females without the syndrome (comparative group). The questionnaire battery was used for data collection (WHOQoL-BREF, Warwick–Edinburgh mental well-being scale, STAI (X-2), and Stress Vulnerability Scale). In processing, comparisons, correlations, and MANCOVA analyses were used. **Results**: The group with tetany syndrome showed significantly lower levels of quality of life (all domains) and well-being and significantly higher anxiousness compared to the group without the syndrome. In vulnerability to stress, a significant difference between groups was not shown. Multivariate testing showed a small interaction effect of tetany syndrome, anxiousness, and stress vulnerability on well-being and quality of life, while anxiousness still had the largest independent effect. **Conclusions**: Lifestyle aspects seem to be a possible intervening factor that, in interaction with anxiety, contributes to a worse quality of life and well-being in individuals with tetany syndrome. The results contribute to the perception of psychological intervention, in terms of stress management and support for a healthy lifestyle, as important in addition to mineral supplementation or medication treatment.

## 1. Introduction

Tetany syndrome is an aggregate of symptoms of increased nervous–muscular system irritability that emerges in and around a cell based on the imbalance and transfer of minerals—calcium and magnesium, as well as sodium and potassium ions. As a consequence of changes in mineral levels, uncomfortable and spasmodic tensions in the muscles manifest [1]. Tetany syndrome manifests in two forms: acute tetany attacks (manifest tetany), or as a latent disease [2]. Among the most frequent symptoms of tetany syndrome, there are muscle cramps, insensitivity, tingling in the extremities, an irregular cardiac rhythm, accelerated breathing, nausea and vomiting, constipation and gassiness, abdominal pains, diarrhea, worsened concentration, weakness, fatigue, and feelings of anxiousness [3]. In the region of Central Europe, especially in the post-communist countries, and probably also in the wake of the crises of the last few years (pandemics; war in Ukraine), which increase the stress load and the demands on anxiety management, psychologists are registering an increase in the diagnosis of tetany syndrome in their clients (which occurred on the recommendation of a general practitioner with a suspicion of an “anxiety syndrome”). Following neurological diagnosis (most commonly a type of EMG examination), and regarding the severity of symptoms, the patient is recommended medication and/or mineral supplementation therapy. Based on the experience of psychologists with these patients, we record a recommendation for psychiatric examination and intervention at most when severe tetanic seizures occur. The latent form, or the form with non-specific symptoms (leg cramps, facial tremors, palpitations, tachycardia or throat pressure, anxiety or depressive symptomatology, attention deficit, impulsivity, etc.), usually results in a recommendation to take magnesium, without any other psychological intervention. Although medical approaches state that the clinical manifestations of stress and magnesium deficiency are virtually identical and most often include fatigue, irritability, anxiety, insomnia, and headaches, supplementation therapy is essential, but as long as long-term stress and its maladaptive coping persist, it is a vicious circle [4]. For these reasons, the attention in recent years has focused on research into the psychological aspects of tetany syndrome to contribute to the emphasis on psychological intervention in the therapy of this disease. As part of the investigation, we would conclude that there are not many sources available that investigate the psychological correlates of either the onset or these phenomena as a consequence of the disease, both domestic and foreign. Based on a review of the literature, we feature the resources most focused on linking the psychological constructs of well-being, quality of life, and disorders that have a link to tetany syndrome based on stress determination or with similar symptomatology. One of them that is important to mention is a panic disorder, which shows a similar clinical picture to that of tetany syndrome due to non-specific manifestations that arise from different mechanisms. In panic disorder, psychological attacks of anxiety or phobia appear without a clear cause, and the psychological picture may not show deviations. In tetany syndrome, there is always a deviation in anxiety and depression, which accompany tetany symptoms, and an attack has similar signs to panic disorder [5]. Up to now, studies have not confirmed with certainty the comorbidity of tetany and panic disorder; however, in previous research [6], 52.4% of respondents had panic disorder diagnosed together with tetany syndrome. The authors found that people with tetany syndrome manifest a higher level of anxiousness. Panic disorder and tetany attacks were identified as significant variables in the level of anxiousness. These findings confirm the research of Majková [7].

Regarding quality of life, Klerman et al. [8] found that individuals with panic disorder rate their emotional and physical health lower than people without the disorder. Carrera et al. [9] found that attacks and agoraphobic avoidance were variables responsible for worsening quality of life in terms of physical functioning and mental health in patients with panic disorder. Kubandová [10] found that people with tetany syndrome achieved worse results in terms of quality of life in the domain of physical health. This is substantiated by the negative influence of physical manifestations caused by tetany. This finding is in accordance with the research of Zbínová [11], where respondents with tetany achieved worse results in the domains of physical and mental health, overall satisfaction with health, and overall quality of life. Barrera and Norton [12] researched the deterioration of quality of life in people with generalized anxiety disorder, social phobia, and panic disorder. The results showed that the individuals with these disorders listed lower satisfaction with the quality of their lives. Müller-Tasch et al. [13] and Altintas et al. [14] point towards panic disorder as a significant factor influencing the quality of life.

In several studies [15,16,17], the vulnerability to stress was found to be substantially higher in people with anxiety and depression disorder. This finding supports the research of Lendić [18], who dealt with perfectionism and disposition to stress in people with tetany syndrome. The analysis discovered that people with tetany have a higher vulnerability to stress. However, it is questionable how the concept of stress disposition or vulnerability is understood. Two concepts can be discerned from studies: vulnerability as a personality trait (in the sense of high sensitivity, disposition, e.g., [15,19]), or as a concept involving lifestyle components such as psycho-hygienic habits or environmental (social, economic, occupational) components [16,20]. In terms of psychological intervention options, the concept of understanding vulnerability to stress as a measure of aspects of lifestyle (in terms of frequency or level) that may support coping or accentuate the experience of stress may be useful for intervening in the development or maintenance of stress-related diseases.

Babinčák et al. [21] researched the predictive power of anxiety as a current state and anxiousness as a personality variable regarding the individual areas of subjectively evaluated quality of life. Persons with low levels of anxiousness reached significantly higher scores in all areas of quality of life in comparison with people with higher levels. Using multivariate hierarchical regression analysis, they found that anxiousness as a personality trait explains 41% of the variability in psychosocial health, 12% for physical health, and 19% for the cognitive area of the subjectively evaluated quality of life. In all of the tested models except for the environmental area, anxiousness was shown to be a stronger predictor of quality of life. Cruz [22] researched the quality of life in patients with Gitelman syndrome, which manifests clinically through tetany spasms and normal or lower-than-normal values of blood pressure [23], similar to tetany syndrome. In the research, 45% of patients with this syndrome considered symptoms such as cramps, muscle weakness and pains, fatigue, overall fatigue, and dizziness to be a big problem, and they showed significantly lower quality of life. Slivková [24] found that people with tetany syndrome have a lower level of mental well-being compared to the non-clinical population, and respondents rated their life satisfaction and work performance after a tetany diagnosis as worsened.

Based on previous knowledge and research, we consider it important to investigate the quality of life and the level of well-being in people with tetany syndrome from the psychological perspective, so that it is possible to find sources of prevention or therapy. The justification for this research is based on the objective lack of studies linking tetany syndrome (as a neurological diagnosis) to psychological causes or consequences (generally psychological correlates and aspects). The cyclical nature of the link between the experience of chronic stress and mineral deficiency justifies the investigation of the interventions to break the cycle, which may lie precisely in psychological intervention. Therefore, it is essential to look for empirical evidence as to which of the above psychological aspects (personality, social-psychological) can be changed by intervention, which is the rationale of the present study.

Quality of life and well-being are understood as health psychology constructs that reflect both mental and physical health based on the individual’s ability to function and adapt in various areas of life, including the environment, work, and relationships. We consider personality disposition anxiousness to be the main trait related to emotionally experiencing tetany symptoms and attacks, and stress vulnerability as a construct that represents a set of factors that increase the risk that the individual will be prone to stress. The aim of this study is to verify anxiousness and stress vulnerability as constructs responsible for lower quality of life and mental well-being in individuals with tetany syndrome, and to explore their individual and multiple effects within the comparison with individuals without tetany syndrome. Based on the overview, it was possible to establish the following hypotheses and one research question:

**H1.** 
*The group with tetany syndrome will show lower well-being, lower quality of life, and higher anxiousness compared to the group without tetany syndrome.*


**H2.** 
*Anxiousness will be correlated with well-being and quality of life in both groups.*


**Q1.** 
*What are the individual effects and combined effects of the presence of tetany syndrome, anxiousness, and vulnerability to stress on well-being and quality of life?*


## 2. Materials and Methods

This work has a non-experimental, comparative–correlational design, using a questionnaire method for data collection. Its comparative character consists in the differentiation of two groups according to the presence of a tetany syndrome (hereinafter abbreviated “TS”): 1—without TS, 2—with TS.

### 2.1. Research Sample

It should be noted that the data and results of this study are part of a larger multi-year research project, and that a national grant funded by a Slovak government agency was obtained based on them, the solution of which is also ongoing currently. The sample of the present research was composed of 267 respondents of female sex, 144 of them with TS (aged 17–63 years; M = 36.08; SD = 10.50) and 123 without TS (aged 17–67 years; M = 34.57; SD = 12.59), from various regions of Slovakia. The research data collection took place between October 2021 and January 2022, which was the end period of the COVID-19 pandemic, and due to ongoing constraints, electronic data collection was chosen. The subjects of the research population were selected through criterion sampling and snowball sampling. For the selection of the target group of women diagnosed with tetany syndrome, the criterion was a finding from a neurological examination, the presence of which was self-reported by the respondents through two items. In one, they were asked whether they had been diagnosed with TS, and based on a positive answer, they continued to the question of when they were diagnosed. In the above question, two respondents chose the option “I don’t know”—we explain the reason for keeping these two respondents in the TS group, including the risk assessment of this decision, in the discussion below.

Given the selection of a specific target sample, for this purpose, groups on online social networks (Facebook) bringing together people with tetany syndrome were identified, and an invitation to participate in the research was shared through these networks. Comparison group subjects, representing the general population, were obtained by opportunity sampling combined with a snowball sampling technique. Students involved in the research project reached out to people in their personal social networks and, following a positive response, distributed informed consent and questionnaire batteries in a targeted manner via electronic communication. The participants could, if they wished, provide additional contacts to other people, and the above process was repeated. These subjects also responded to the question regarding the diagnosis of tetany syndrome by the findings of a neurological examination and, in case of a positive answer, were included in the target research group (with TS). In both groups, the exclusion criterion was the presence of a history of psychiatric, neurological, or other serious illness or intervention (e.g., cancer, organ transplantation, amputation, kidney disease with dialysis, etc.). Protocols with missing data were also excluded. In the group with TS, several aspects related to the disease were detected, a description of which can be seen in Table 1.

### 2.2. Measurements

A questionnaire battery consisting of the following instruments was used for data collection: The WHOQOL-BREF questionnaire (WHO official Slovak version), which measures quality of life (QoL) made up of four domains—physical health, psychological health, social relationships, and environment—on the basis of 26 items [25]. The Warwick–Edinburgh mental well-being scale (WEMWBS) consists of 14 items to measure aspects of well-being (WB), including positive thinking, satisfaction with interpersonal relationships, positive functioning, etc. [26] (Tennant et al., 2007). The STAI X-2 (standardized Slovak version by Müllner et al.) [27] questionnaire for anxiousness (ANX) measurement contains 20 items, which a respondent evaluates on a four-point Likert scale based on how they usually feel. The Stress Vulnerability Scale (SVS) by Miller and Smith [28] measures how prone a person is to physical and psychological stress (SV). The scale contains 20 items, which are evaluated on a five-point Likert scale. The questionnaire battery was supplemented by questions concerning tetany syndrome, such as its diagnosis, the presence of panic disorder, the occurrence of symptoms and attacks, other serious diagnoses, etc. Informed consent was obtained by the researcher from all participants before completing the questionnaire.

In the case of the SVS and WEMWBS tools, the original Slovak versions were created based on a professional translation validated by experts and psychometrically verified in a pre-survey according to the rules of psychological non-clinical research methods. All included variables were of the scale type, with a higher score indicating a higher intensity or level of manifestation of phenomena. The Cronbach’s alpha values for the questionnaires used in the research are as follows (N = 267): physical health α = 0.699; psychological health α = 0.829; social relationships α = 0.691; environment α = 0.752, WB α = 0.834; ANX α = 0.928; SV α = 0.804.

### 2.3. Data Processing Methods

For the analysis of the data, the IBM SPSS Statistics 26 software was used. All scale variables showed a normal distribution in the groups compared (based on the Shapiro–Wilk test, *p* > 0.05). Student’s *t*-test was applied for the exploration of group differences, and Pearson’s correlation was used for interrelations. Using multivariate analysis of covariance (MANCOVA), we investigated the effects of the factor variables and covariates on the two dependent variables. Testing analyses were supplemented by description and visual aids.

## 3. Results

Firstly, the differences in variables between the groups with and without TS were examined (Table 2). The analysis showed statistically significant differences (*p* < 0.01) in all domains and the global score of QoL, as well as in WB and ANX. Based on the effect sizes, we interpreted the intermediate effects of TS on higher anxiousness (η^2^ = 0.105), lower global QoL (η^2^ = 0.065) and physical health domain (η^2^ = 0.077), and small effects on lower QoL domains: psychological health (η^2^ = 0.034), social relationships (0.041), environment (η^2^ = 0.019), and WB (η^2^ = 0.045). There was no significance in the comparison of SV (*p* > 0.05). An illustration of the mean values in the compared groups can be found in Figure 1.

The next step of the analysis was a calculation of intercorrelations between the investigated variables for each group separately (with and without TS). Table 3 shows the results, on the basis of which we interpreted all detected correlations as significant (*p* < 0.05), from weak to very strong relations (in the group without TS: ±0.195 to ±0.881; with TS: ±0.256 to ±0.837). But it should be noted that some relations are visibly different in strength between groups. In Table 2, the r values that differ between groups are highlighted. These include almost all correlations with SV, except for the social relationship QoL domain, and a relationship between ANX and the environment QoL domain, which are stronger (applying transformation with Fisher’s Z for two correlations) [29] in the group with TS than in the group without TS. On the other hand, the correlation of different values for the pair of social relations and psychological health is higher in the group with TS.

The differences in the correlations detected between groups and the strong association between WB and QoL suggested the construction of a multivariate model, in which we tested the individual and interaction effects of SV, ANX, and TS (with/without groups) on WB and QoL within MANCOVA. The conditions of equality of covariance matrices and error variances (Box’s test and Levene’s test with *p* > 0.05) were evaluated before applying the calculation. As the results show (Table 4), there was a significant (*p* < 0.05) individual small effect of the presence of TS (partial η^2^ = 0.025) and a large effect of ANX (partial η^2^ = 0.219). Furthermore, in the interactions, we interpreted the significant small effects of the presence of TS with ANX (partial η^2^ = 0.023), but also of TS with SV (partial η^2^ = 0.028) and the interaction effect of all three factors (TS * ANX * SV; partial η^2^ = 0.028), on two related dependent variables: WB and QoL. We identified no significant (*p* > 0.05) individual effect of a covariate SV or interaction effect of SV with ANX on the dependent variables.

## 4. Discussion

The goal of our work was to research the quality of life and well-being levels in people with tetany syndrome and determine their levels of anxiousness as a personality trait and vulnerability to stress as a lifestyle indicator. The statistical analysis showed a significant difference in all domains of quality of life, and also in well-being. With regard to average scores and Czech population standards [30], we consider the differences in physical and psychological health, in which respondents with TS scored markedly lower than respondents without TS, to be the most significant. The findings confirm Hypothesis 1, in accordance with research on the Slovak population with tetany syndrome [10,11]. The worsened QoL and WB in respondents with tetany are substantiated by the uncomfortable symptoms that accompany tetany. The findings match those of Cruz [22], wherein 45% of patients considered symptoms such as spasms, muscular weakness and pain, fatigue, and dizziness to be a big problem. These patients also showed a significantly lower level of quality of life. These findings match those of Tušek-Bunc and Petek [31], Chupáčová et al. [32], Zbínová [11], and Slivková [24], wherein respondents with tetany syndrome showed lowered levels of QoL and mental well-being. We further targeted the difference in anxiousness between respondents with and without tetany. The analysis showed a difference, which confirmed the results of previous research [6,7]. In vulnerability to stress, in the comparison between respondents with tetany syndrome and without, a difference was not shown, which is not in accordance with the results of Lendić [18]. According to Weger and Sandi [33], very anxious individuals show changes in behavior and cognitive deficits together with more reactive physiological stress reactions, and they consider anxiety to be a significant factor in stress vulnerability. Previous findings show that people who are significantly more anxious will show a higher disposition to stress. According to Müllner et al. [28], anxious people have a tendency to perceive situations and conditions containing a chance of failure with a higher intensity. Based on our results, we interpreted only a weak association of anxiety and vulnerability to stress in the non-TS group, and a strong association in the TS group. We argue that the reason for this lies in the construct of TS as a lifestyle variable that does not determine the degree to which stress is experienced but can be considered as a circumstance that may exacerbate or, conversely, worsen coping, so it is generally not directly related to anxiousness.

As part of testing the second hypothesis, the results supporting the hypothesized relationship between anxiety and quality of life or well-being were found; however, we can interpret the different strengths of the connections between groups. Babinčák et al. [21] researched the predictive power of anxiousness as a personality variable with regard to individual areas of subjectively evaluated quality of life. They found that anxiousness explained 41% of the variability in values in the psychosocial area, 12% in the physical area, and 19% in the cognitive area. Anxiousness was shown to be a strong predictor of QoL values and WB [34]. Concerning the findings stating that respondents with TS are more anxious, have worse well-being and quality of life, and at the same time do not differ in vulnerability to stress from respondents without TS, we decided to include anxiousness, vulnerability to stress, and the group factor (TS, non-TS) in the multifactorial model and clarify the individual and combined influences of these factors on both quality of life and well-being (as markers of general health and functioning) more specifically. The results of the multivariate analysis identified anxiousness as the strongest predictor of QoL and WB. These findings are in contrast with those of Carrera et al. [9] or Altintas et al. [14], who identified attacks and agoraphobic avoidance as the variables corresponding to deteriorating quality of life in terms of physical functioning. Based on the results put forward, we can state that respondents with tetany syndrome have significantly lowered levels of physical and psychological health as well as mental well-being, and they are significantly more anxious. In answering Q1, we can interpret the relationship between anxiety and general functioning and health regardless of the presence of a TS diagnosis. Similarly, the interaction effect of anxiety and vulnerability to stress on quality of life and well-being was observed in both groups but was significantly stronger in the tetany group than in the comparison group. We can summarize that people with TS are more anxious, and that all of their conditions and everyday habits are similar (in terms of lifestyle understanding of vulnerability to stress), which is associated with a lower quality of life and mental well-being than would be the case for people without a diagnosis. The mental processes caused by a TS diagnosis, whether due to mineral deficiency or nervous system irritability, accentuate the effects of anxiety on normal functioning or perceived health in the context of environmental conditions and daily habits.

### 4.1. Limitations

In this research, we can mention several aspects that did not take place according to the original research plan. The first limit is the collection of data during the end period of the COVID-19 pandemic, which applies to both groups and does not assume distortion. The second limit is the online snowball technique for obtaining the sample, which was justified due to the specificity of the population. The online form of data collection may also cast doubt on whether the respondents’ self-reported diagnosis of TS was truthful. In the country where the research was carried out, and in the wider region, the diagnosis of TS is so little known that for many people it is the first time they have heard of it when they are diagnosed. For this reason, although possible, we believe that it is unlikely that the respondents distorted this information. The research team also sought to support the validity of the data by targeting the questionnaire to certain individuals, and it was not a freely available form on the Internet. By not making the form publicly available to random individuals, the collection was partially protected from respondents interested in completing various surveys on the Internet, in both research groups. The self-reporting of the syndrome without confirmation by a practitioner is another limitation of this study, especially in the comparison group, where we do not know whether the diagnosis of TS would be confirmed or not. However, we worked with the idea that awareness of the diagnosis may be an essential factor in managing stress and anxiety in people with TS. Collaboration with neurologists and the implementation of a clinical study that would remove this limitation are incentives for further research.

To some extent, the inclusion of the two respondents who reported “I don’t know” in response to the question about when they were diagnosed with the syndrome may be questionable. Responses included options up to a maximum of “6 years or more”, and there was also a choice of “Don’t know”. We did not exclude these respondents from the TS group (or from the research), because we explained their answers on the basis that the diagnosis was so long ago that they did not remember, and therefore made that decision. Considering that this was the second question regarding the TS diagnosis, we do not perceive this solution as risky, especially considering the sample size and the minimal possibility of impact (in case of error) on the statistical results and substantive interpretations. There may be a limit to the use of the questionnaire as a self-report method, which always carries risk, but there is no alternative for investigating psychological constructs. Therefore, we tried to eliminate systematic errors in the measurement instruments. In particular, validation studies of Slovak versions are not available for two instruments (measuring well-being and stress vulnerability), so we tried to ensure basic properties (validity, reliability, comprehensibility) for research purposes, but without population norms. Thus, a disadvantage of this study is the impossibility of determining the levels of the constructs under study in comparison with a population reference.

### 4.2. Practical Application

Accentuated traits are part of risk potential, and knowing them can contribute to increased psychological self-care. The description of psychological correlates and predictors of the disease will allow, in addition to pharmacological treatment, for the determination of the appropriate psychological intervention for the patient. The results of this study draw attention to individual tendencies of experiencing life situations, the importance of emotional states, lifestyle changes, and activity, and the importance of psychotherapy in the intervention after the diagnosis of the syndrome.

The results may contribute to the elimination of negative attitudes to the need for psychotherapeutic intervention, which are specific to conservative peoples in post-communist European countries and point to its importance in favor of a favorable mental health adjustment in the case of tetany syndrome. In the absence of comprehensive research on the psychological variables of tetany syndrome, we consider it highly beneficial to extend the theoretical and empirical framework to the issues that are partially raised by the results of this study. The results may be a stimulus for qualitative or longitudinal investigation of the causal chain between anxiousness and tetany symptomatology, where anxiousness as a personality trait would be considered a risk factor, but anxiety symptoms may also be a consequence of previous experience with tetany attacks. It is also possible to consider and verify the mediation effect of risky lifestyle aspects on the link between anxiousness (or neurotic personality platform) and tetany syndrome within longitudinal research. In a broader sense, this notion can be extended to include the experience of tetany attacks as a trigger for the occurrence of panic attacks.

The broader application has the prospect of compiling the findings (together with other results of the broader project) into an information booklet containing recommendations, measures, and interventions, as well as information on sources of help that would be available in clinical diagnostic units and neurological outpatient clinics in order to motivate persons diagnosed with tetany syndrome to undertake psychological intervention in terms of promoting positive aspects of personality, reducing resilience to stress, coping with anxiety, and promoting the development of adaptive coping strategies and changes in lifestyle components.

## 5. Conclusions

Respondents with TS have a lowered quality of life and well-being. The findings of previous research were confirmed, in that respondents with tetany were significantly more anxious, but there was no difference shown between the respondents with tetany syndrome and the rest of the population in terms of vulnerability to stress. Anxiousness was shown to be an important factor corresponding to the lowered levels of quality of life and well-being, but concerning TS, stress vulnerability seemed to be an intervening factor that worsened the quality of life and mental well-being in people with this diagnosis.

## Figures and Tables

**Figure 1 brainsci-15-00358-f001:**
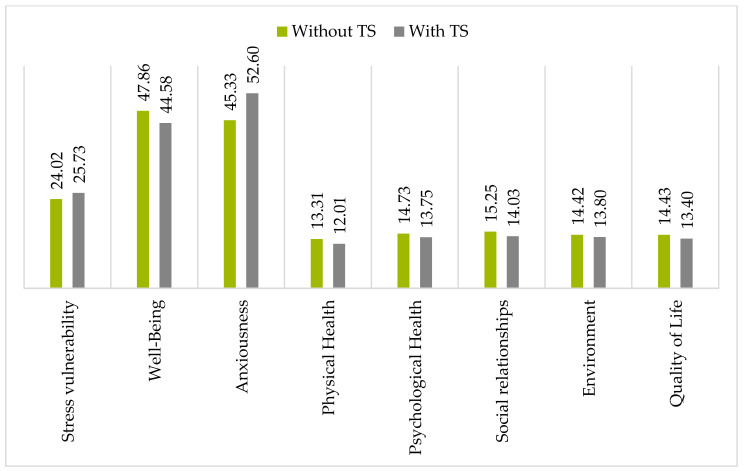
Illustration of mean values of variables in the two groups compared.

**Table 1 brainsci-15-00358-t001:** Research sample description.

		Without TS		With TS		Total	
		*n*	%	*n*	%	*n*	%
Age	Under 30	60	48.8	45	31.3	105	39.3
	30–39 years	19	15.4	39	27.1	58	21.7
	40–49 years	29	23.6	40	27.8	69	25.8
	Over 50	15	12.2	20	13.9	35	13.1
Education	Primary	5	4.1	3	2.1	8	3
	Secondary (specialized)	29	23.6	10	6.9	39	14.6
	Complete secondary (with the graduation)	53	43.1	66	45.8	119	44.6
	Higher, 1st. BA	16	13	23	16	39	14.6
	Higher, 2nd. MA	16	13	40	27.8	56	21
	Postgraduate, 3rd. PhD.	4	3.3	2	1.4	6	2.2
Health	Bad	3	2.4	16	11.1	19	7.1
	Nor good nor bad	22	17.9	56	38.9	78	29.2
	Good	74	60.2	64	44.4	138	51.7
	Very good	24	19.5	8	5.6	32	12
Time since the diagnosis of TS	I don’t know			2	1.4		
	Less than 2 years			52	36.1		
	3–5 years			44	30.6		
	6 and more years			46	31.9		
Symptoms of TS	Yes			82	56.9		
	No			41	28.5		
	I don’t know			7	4.9		
	Sometimes			14	9.7		
Panic disorder diagnosis	No			92	63.9		
	After the diagnosis of TS			28	19.4		
	Before the diagnosis of TS			13	9		
	Together			7	4.9		
	It was not examined			4	2.8		
Frequency of attacks	Once and more time in a week			25	17.4		
	Once and several times a month			33	22.9		
	Several times during the half-year			21	14.6		
	Several times during the year			42	29.2		
	Almost never			23	16		
Pharmacotherapy of TS	No			72	50		
	Yes			72	50		

**Table 2 brainsci-15-00358-t002:** Comparison of variables between groups: without TS (n = 123) and with TS (n = 144).

Variables	Group							
	Without TS		With TS		Student’s *t*-Test		Effect Sizes	
	M	SD	M	SD	*t*	MD	Cohen’s d	η^2^
QoL (global)	14.43	1.92	13.40	2	4.265 **	1.03	0.525	0.065
Physical health	13.31	2.32	12.01	2.17	4.713 **	1.30	0.580	0.077
Psychological health	14.73	2.58	13.75	2.64	3.027 **	0.97	0.375	0.034
Social relationships	15.25	2.66	14.03	3.17	3.387 **	1.22	0.414	0.041
Environment	14.42	2.14	13.80	2.27	2.298 **	0.62	0.280	0.019
WB	47.86	7.37	44.58	7.68	3.551 **	3.29	0.436	0.045
ANX	45.33	10.35	52.60	10.86	−5.579 **	−7.28	0.685	0.105
SV	25.73	10.20	25.73	11.59	−1.271	−1.71	0.156	0.006

Note: M—mean; SD—standard deviation; MD—mean difference; **—*p* < 0.01.

**Table 3 brainsci-15-00358-t003:** Correlations (Pearson’s r) for research variables in two groups (with and without TS).

Group/Variable	1	2	3	4	5	6	7	8
Without TS								
1. SV	-	-	-	-	-	-	-	-
2. ANX	**0.195 ***	-	-	-	-	-	-	-
3. WB	**−0.274 ****	−0.709 **	-	-	-	-	-	-
4. Physical health	**−0.266 ****	−0.540 **	0.466 **	-	-	-	-	-
5. Psychological health	**−0.214 ****	−0.678 **	0.716 **	0.608 **	-	-	-	-
6. Social relationships	−0.369 **	−0.456 **	0.536 **	0.334 **	**0.626 ****	-	-	-
7. Environment	**−0.273 ****	**−0.484 ****	0.472 **	0.458 **	0.563 **	0.572 **	-	-
8. QoL (global)	**−0.356 ****	−0.684 **	0.699 **	0.738 **	0.881 **	0.800 **	0.792 **	-
With TS								
1. SV	-	-	-	-	-	-	-	-
2. ANX	**0.579 ****	-	-	-	-	-	-	-
3. WB	**−0.537 ****	−0.757 **	-	-	-	-	-	-
4. Physical health	**−0.514 ****	−0.616 **	0.545 **	-	-	-	-	-
5. Psychological health	**−0.468 ****	−0.766 **	0.789 **	0.568 **	-	-	-	-
6. Social relationships	−0.447 **	−0.452 **	0.508 **	0.256 **	**0.429 ****	-	-	-
7. Environment	**−0.542 ****	**−0.623 ****	0.598 **	0.517 **	0.601 **	0.545 **	-	-
8. QoL (global)	**−0.624 ****	−0.774 **	0.778 **	0.706 **	0.823 **	0.760 **	0.837 **	-

Note: * = *p* < 0.05, ** = *p* < 0.01.

**Table 4 brainsci-15-00358-t004:** Results of between-subject effects and multivariate testing by MANCOVA.

Source	Multivariate Test ^a^			Test of Between-Subject Effects		
	Value	F	Partial η^2^	Dependent	F	Partial η^2^
Intercept	0.216	468.132 **	0.784	WB ^b^	584.470 **	0.587
				QoL ^c^	719.083 **	0.628
TS (Group)	0.975	3.347 *	0.025	WB	4.251 *	0.016
				QoL	5.097 *	0.019
ANX	0.781	36.184 **	0.219	WB	58.309 **	0.184
				QoL	42.165 **	0.140
SV	0.998	0.302	0.002	WB	0.598	0.002
				QoL	0.151	0.001
TS * ANX	0.977	3.087 *	0.023	WB	4.271 *	0.016
				QoL	4.366 *	0.017
TS * SV	0.972	3.664 *	0.028	WB	5.122 *	0.019
				QoL	5.138 *	0.019
ANX * SV	0.972	0.567	0.004	WB	0.029	0.001
				QoL	0.811	0.003
TS * ANX * SV	0.972	3.648 *	0.028	WB	5.249 *	0.020
				QoL	4.964 *	0.019

Note: * = *p* < 0.05, ** = *p* < 0.01; ^a^: Wilks’ λ. Dependent: well-being * quality of life—Box’s test: *p* > 0.05. ^b^: Dependent: well-being—R squared = 0.587 (adjusted R squared = 0.576). ^c^: Dependent: quality of life—R squared = 0.628 (adjusted R squared = 0.617).

## Data Availability

The data presented in this study are available upon request from the corresponding author due to ethical reasons (the participants provided personal data about their health, and their disclosure was not foreseen in the informed consent).

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
