# Peer review of "Quality of Life and Well-Being in Women with Tetany Syndrome in the Context of Anxiousness and Stress Vulnerability"

_brainsci, 2025, doi:10.3390/brainsci15040358_

Round 1

Reviewer 1 Report

Comments and Suggestions for Authors

I would like to express my gratitude to the editors and authors for the opportunity to review the article "Quality of Life and Well-being in Women with Tetany Syndrome in the Context of Anxiousness and Stress Vulnerability." The article presents a relevant and well-founded approach to the relationship between quality of life, well-being, and psychological factors in women with tetany syndrome. The topic is pertinent and has the potential to provide valuable contributions to the understanding of the impacts of this condition on both mental and physical health. With some improvements in the structure and clarity of the text, particularly in the presentation of results, the deepening of the discussion, and the strengthening of the methodological justification, the article has the potential to make a meaningful contribution to the field of mental health and quality of life. Below, I present my contributions and suggestions for improving the manuscript:

  1. Abstract: The abstract clearly presents the objectives, methodology, and results, but the conclusion could better emphasize the practical relevance of the findings, highlighting their potential implications for research and clinical practice. 2.Introduction The introduction provides a good contextualization of Tetany Syndrome (TS) but could be more concise, reducing redundancy in the symptomatology description. The link between tetany and anxiety is well-supported, yet the concept of stress vulnerability needs clearer definition. Adding a clear hypothesis at the end would strengthen the introduction by explicitly stating the study's expected outcomes based on previous research.

2. Materials and Methods In the article, the participant recruitment period is not explicitly mentioned. To ensure transparency and methodological rigor, it is recommended that the authors specify the exact dates of data collection, clearly indicating when the study was conducted.

Furthermore, it appears that data may have been collected before the ethics committee's approval ("The limit is the collection of data during the COVID pandemic" - line 223), which raises a critical ethical concern. Data collection prior to ethical approval should not occur, as it compromises research ethics principles and participant protection. It is strongly recommended that the authors clarify this issue and ensure that all ethical procedures were followed in accordance with international standards.

The methods section is well-structured; however, it would be important to make the inclusion and exclusion criteria more explicit, clearly defining the parameters used for sample selection. The article mentions some potential sources of bias, but this discussion could be further developed. For instance, recruitment was conducted through social media groups, which may have resulted in a non-representative sample of the general population with tetany syndrome, limiting the generalizability of the findings. Additionally, the self-administered questionnaires used in the study may introduce response bias, as the data collected relies exclusively on participants' subjective perceptions. However, the article does not present any strategies to mitigate this bias.

Another important aspect to clarify is the sample size determination. The authors should describe the rationale used to determine the number of participants, specifying whether it was based on power calculations or predefined criteria. While the choice of questionnaires is appropriate, a stronger justification would be beneficial, particularly for instruments with Cronbach's Alpha below 0.7, whose reliability should be further discussed. Regarding statistical analyses, the explanation of MANCOVA could be more detailed, specifying whether the assumptions of normality, homoscedasticity, and independence of errors were verified and what measures were taken if these assumptions were not met.

It would also be important for the article to explain how missing data was handled, as well as the procedures adopted to ensure participant data protection, including when and how these data will be destroyed. Lastly, it is recommended that the authors follow the STROBE Statement-checklist guidelines, ensuring that all essential methodological aspects of observational studies are thoroughly described and addressed.

3. Results The results section clearly presents the differences between the groups, but the interpretation of effect magnitudes could be further elaborated. Additionally, exploring moderator or mediator variables would provide a more comprehensive analysis of the results.

4. Discussion The discussion is coherent but could more directly relate the results to the literature. The absence of differences in stress vulnerability requires a more in-depth analysis. It would also be relevant to address the clinical implications of the results and suggest directions for future research. Additionally, the use of online snowball sampling, while valid, may introduce selection bias, and this limitation should be discussed in greater depth.

5. Conclusions The conclusion summarizes the main findings of the study but could be strengthened by including a more targeted recommendation for clinical interventions or health policies, highlighting the practical implications of the results. References The article provides a solid bibliographic foundation, but some references appear to be undergraduate theses, which may weaken its scientific rigor. Prioritizing peer-reviewed studies would enhance credibility. Additionally, only 11 out of 29 references are less than five years old. Updating the references with more recent studies would strengthen the work."

The manuscript requires major revisions.

Author Response

Dear Reviewer, let me first thank you for your comments on the study.
We sent a statement on ethical practices in the first instance. I we are now sending a statement regarding the other comments.

  1. Abstract: Edited according to all comments- the concept of stress vulnerability defined and new literature sources added
  2. Materials and Methods:
    Psychometrics
    Slovak versions of the questionnaires were used for data collection, which underwent a standard process of verification of psychometric characteristics. Given the specialisation of both authors in the field of psychometrics (creation and validation of instruments for the Slovak population), method verification is considered a basic and key step in the research. In the case of the WHOQOL-BREF questionnaire, the official version available online on the WHO website was used, and Cronbach's alpha was verified on the research sample. For the SVS and WEMWBS tools, the original Slovak versions were created based on a professional translation validated by experts and psychometrically verified in a pre-survey according to the rules of psychological non-clinical research methods. In the framework of the study, we present the reliability values as internal consistency, while the other parameters are not the subject of the study itself and we do not see relevance in their inclusion. We have emphasised this in the article.
    We have included the limitations resulting from the use of the self-report method, online collection, snowball sampling, etc. in the article under the limitations section.

Statistical Analyses
The explanation of MANCOVA has been added to the article.

Research Sample Description
Given that relevant data on the prevalence of the disease in the population are not available in Slovakia, we determined the sample size based on statistical theoretical assumptions to a minimum number of 100 respondents.
As part of the description of the research sample, we have added data regarding the collection period and criteria for inclusion in the research, based on comments.
These subjects also responded to the question regarding the diagnosis of tetany syndrome according to the findings of a neurological examination, and in the case of a positive answer, they were included in the target research group (with TS). In both groups, the exclusion criterion was the presence of a history of psychiatric, neurological, or other serious illness or intervention (e.g. cancer, organ transplantation, amputation, kidney disease with dialysis, etc.). Protocols with missing data were also excluded.

The STROBE Statement-checklist is not used in psychological research. The research is of a psychological nature with applications to the field of neurology and follows the rules of psychological research. The research sample consists of a group with a neurological diagnosis; however, the subject of the research is psychologically significant constructs.

Discussion and Conclusions:
Literature Sources and Research Justification
The use of bachelor’s and master’s theses as sources is justified precisely by the lack of peer-reviewed studies addressing the psychological constructs of tetanic syndrome. This highlights the underestimation of their investigation within the context of the presented study. Similarly, there are no more recent data. Most sources approach the syndrome from a medical perspective.

The Practical Relevance of the Findings and Recommendations for Clinical Interventions:
This was added to the article, including a more consistent description of the limitations.

Authors‘ explanation about the ethic approval is as below, and the document is attached.

In Slovakia, it was not mandatory for a university to have an ethics committee in the past, and our faculty committee was established only after the COVID-19 pandemic. Therefore, we had the research project, which is long-term and builds on previously implemented collections and results of student work, approved as soon as it was approved for us at the national level (by VEGA agency). When approving VEGA (funding Agency) grants, the project's ethical dimension is also considered. After the grant was approved by the national agency VEGA, we forwarded the project to our existing faculty committee, which approved it.

It is truth, that we did not state precisely in the work that the data we are processing were collected earlier. It was as part of a student's final thesis under our supervision.

But student research is ethically approved by the department management and is supervised by the supervisor, in addition, informed consent was secured from each participant, which we have already documented.

Best regard

Authors

Reviewer 2 Report

Comments and Suggestions for Authors

The manuscript entitled “Quality of Life and Well-being in Women with the Tetany Syn-2 drome in the Context of Anxiousness and Stress Vulnerability” deals with quality of life (QoL), mental well-being (WB), anxiousness, and stress vulnerability in Slovak women with tetany syndrome (TS) comparison with the population without the syndrome. A questionnaire battery was used composed of: WHOQoL-BREF, Warwick-Edinburgh mental well-being scale, STAI 15 (X-2), and Stress Vulnerability Scale. Findings suggest that lifestyle aspects seem to be a possible intervening factor that, in interaction with anxiety, contributes to a worse quality of life and well-being in individuals with Tetany syndrome.

The manuscript can be of potential interest to the Brain Sciences readership but, in its current form, it shows many concerns, which I will enumerate below.

Introduction

The introduction is focused on “Tetany syndrome” but the ICD-10 classification for Tetany is under the range of symptoms, signs, and abnormal clinical and laboratory findings, not elsewhere classified (code R29. 0).

The conceptual foundation of the present study needs to be detailed with a strong analysis of the literature. The background is not sufficiently detailed, and the relevance to the specific research questions could be explicitly discussed paying particular attention to the specific population. I noted that 50% of reported references is in Slovak. Is there a reason? Is it possible to think that the Slovak population has some predisposition or geographical diversity compared to other world areas?

The considerations and reported literature in the literature analysis lack appropriate inquiry that is then able to suggest and derive purpose and specific hypotheses. The connection between cited studies and current research could be clearer. Explicitly linking how each previous review informs or contrasts with the current study’s approach would enhance coherence.

I generally think that the manuscript is poorly written, the methods are not sound and there is a need for editing, especially around the framing, and clarity of methods, tables, and results.

In practice, considering the weak structure of the work, a new study is not justified, and it is not known what this study would add to current knowledge.

Hypotheses should be detailed. The current manuscript contains only an aim (lines 94-98).

Methods

The methods section is not easily readable. No mention is made concerning the crucial aspect of participants’ enrollment, length of it, inclusion/exclusion criteria. Recruitment is not described. This is an important point. Because the recruitment took place as an internal process, results can be biased. Given that the sample was convenience-based, and participation was voluntary, there is a potential for selection bias.

- research sample: the characteristics of the sample are a result and should be included in the results section.

- material (instrument, psychometric tools). It would be imperative to use questionnaire validations and corresponding cut-off scores for the nationality included in the study. I believe the authors found this variable to be of little importance, considering that they wrote that:

“An original translated and psychometrically verified Slovak versions of questionnaires were used”. However, the validation of the psychometric properties of a tool employs several methods and reliability, responsiveness, and stability analysis which I don't think were followed since they are not reported and described. Consequently, some assessments can yield inaccurate results. We use various tools daily for clinical assessments and evaluations, measuring change over time and establishing prognosis for patients. Our clinical reasoning, intervention, and research suggestions can only be as strong as the tools we use. Thus, I believe it is the most serious problem of the present study.

Because Figure 1 reported the same scores shown in Table 2, it is not useful.

Author Response

Dear Reviewer, let me first thank you for your comments on the study.
The response to your revision is at the attachment.

Best regard

Authors

Reviewer 3 Report

Comments and Suggestions for Authors

Please find the comments:

  1. Please proofread the paper as there are obvious errors. For instance, "The paper deals with quality of life (QoL), mental well-being (WB), anxiousness and stress vulnerability in women with tetany syndrome (TS) in comparison with the population without the syndrome, and to investigate individual or interact effects of anxiousness, stress vulnerability and Tetany syndrome diagnosis on quality of life and well-being in Slovak women."
  2. What is the reason of inclusions the phrase "in terms of sex" here? "The research sample was composed of 144 female (in terms of sex) respondents with the diagnosis of Tetany syndrome and 123 without the syndrome (comparative group)." Please add females after 123.
  3. "Warwick-Edinburgh mental well-being scale," Capital letters should be used.
  4. "binary comparative" What is it? This term seems to be used incorrectly.
  5. "Multivariate testing showed a small interaction effect of Tetany syndrome, anxiousness and stress vulnerability on Well-being and Quality of life while anxiousness still has the largest independent effect.". Why were several terms written with capital letters? Please correct English according to its rules.
  6. I am not very familiar of the Tetany syndrome, and I hope that other reviewers will help in assessing descriptions of Tetany syndrome in this paper. In my view, there is a need to shorten the description of the panic disorder and its association with the Tetany syndrome in the introduction. Please focus on the variables of interest and provide a more smoother narration in the introduction.
  7. The relevance of this study should be clearly stated. It is obvious that the QoL and well-being levels are lower in the clinical compared to non-clinical one. Please demonstrate why this study is necessary for clinical practice, and what research gaps it can fullfil.  
  8. Methodology: In my view, the methodology lack clinical precision. This study uses an online sample. It is unclear how the diagnosis was determined. In Table 1, the authors indicated that 2 people did not know whether they had the diagnose of TS, but these people were included in the clinical group.
  9. The description of the recruitment is poor: "get group were women with diagnosed tetany syndrome. The respondents were contacted mainly through groups on social media oriented toward tetany syndrome, given the COVID-19 pandemic. ". How were participants recruited? What social media groups were used? When was the study conducted? How the diagnosis was confirmed? By whom? Based on what?

In my view, the paper lacks rigorous clinical methodology. Readers do not know exactly whether these people were with TS or not. Inclusion 2 people who do not know whether they had a diagnosis to the clinical group implies significant methodological violations in this research. 

It seems that other important clinical variables (e.g., psychiatric disorders), which could be important for this study, were not controlled for. 

Comments on the Quality of English Language

Indicated in the review form.

Author Response

(The authors gave the same response as above.)

Reviewer 4 Report

Comments and Suggestions for Authors

Evaluation of the manuscript “Quality of Life and Well-being in Women with Tetany Syndrome in the Context of Anxiousness and Stress Vulnerability,” submitted to the journal Brain Sciences.

The authors investigated the quality of life and mental well-being of Slovak women with tetany syndrome (TS), comparing them to a group without the condition. The results showed that women with TS had significantly lower levels of quality of life and well-being, as well as higher anxiety levels, although vulnerability to stress did not differ between the groups. Anxiety emerged as the most influential factor in worsening quality of life and well-being, suggesting that lifestyle aspects and emotional factors may exacerbate the syndrome’s impact.

The title adequately reflects the study’s objective, being clear and informative. The abstract is well-structured, covering the objectives, methodology, main results, and conclusions. The introduction effectively contextualizes the topic, highlighting its relevance and reviewing the pertinent literature.

The study employs a non-experimental comparative-correlational design, using questionnaires for data collection. The sample was selected through convenience sampling, with recruitment via social networks and the “snowball” technique. While this approach is understandable given the constraints imposed by the COVID-19 pandemic, it may introduce selection bias, potentially limiting the generalizability of the results. The choice of instruments is appropriate for measuring quality of life (WHOQoL-BREF), mental well-being (WEMWBS), anxiety (STAI X-2), and vulnerability to stress (SVS). However, the lack of formal validation of the Slovak translations may affect data reliability, a limitation acknowledged in the discussion.

The statistical analyses were well-executed, including Student’s t-tests, Pearson’s correlations, and MANCOVA to examine individual effects and interactions between variables. The approach is robust and supports valid inferences within the sampling limitations, particularly regarding the differences observed between the groups. However, the discussion could better highlight the practical implications of the findings, avoiding an overly statistical interpretation.

The results are presented clearly and in a well-structured manner, with statistically significant differences discussed in the context of existing literature. However, it is important to more clearly distinguish correlation from causation to prevent misinterpretation. Additionally, a deeper reflection on the practical applicability of the findings would enhance the study’s relevance, particularly by exploring potential implications for clinical interventions and preventive strategies.

Overall, the study addresses a relevant and well-grounded topic with a solid methodology. However, certain limitations should be considered, such as sampling bias, the lack of formal validation of the instruments in the Slovak population, and the need for greater clarity in discussing the findings.

I suggest that the authors:

Clarify the distinction between correlation and causation in the discussion of results;

Expand the discussion on the practical applicability of the findings;

Provide a more in-depth analysis of methodological limitations, particularly concerning sampling and instrument validity.

Author Response

(The authors gave the same response as above.)

Round 2

Reviewer 1 Report

Comments and Suggestions for Authors

I would like to express my gratitude for the opportunity to review the manuscript “Quality of Life and Well-being in Women with Tetany Syndrome in the Context of Anxiousness and Stress Vulnerability.”

I appreciate the authors' efforts to address the suggestions provided in the initial review. The clarification regarding ethical approval — including reference to the Declaration of Helsinki, departmental oversight in 2021 (as part of a student thesis), and institutional ethics committee approval in July 2024 — is noted and appears appropriate. However, according to international standards, ethics committee approval should be obtained prior to any data collection. This point remains a concern and should be carefully considered in the context of research ethics compliance.

Overall, the manuscript presents a relevant and thoughtful contribution to the understanding of psychological factors associated with tetany syndrome. With the improvements made in response to the initial feedback, I believe the article can offer valuable insights in the fields of mental health and quality of life.

Final Decision:
Accept for publication.

Thank you for the opportunity to contribute to the review process.

Kind regards,

Author Response

Dear reviewer,
We sincerely thank you for your positive opinion and for your positive assessment of our efforts to revise the paper. We know that the first version had many shortcomings (even though we found errors in the interpretations of the results in the discussion, which we corrected), and thanks to your comments, the article became more valuable and worthy of publication. We greatly appreciate this opportunity.
At the same time, we are aware of the shortcomings in the ethical review procedures, which was a major problem in Slovakia a few years ago. We would like to explain this to you.
Since we have been working in an academic space, there have been no ethics committees at our university or even centrally in Slovakia (except for the ethics committee for practicing psychologists). With national grants, obtaining them was a guarantee that ethical standards were also taken into account during the approval process. If we wanted to publish research in foreign journals, documenting ethical approval was a problem. Therefore, our team initiated the establishment of an ethics committee at the faculty, which has been working for several years (since 2022). However, student theses, of which we have over 200 per year in psychology, cannot be approved by the committee, and therefore the supervisor and head of the department are responsible for ethical implementation.
As teachers responsible for teaching methodological subjects, we train students on how to ensure all ethical principles, we prioritize ensuring informed consent, the rights to safety, and dignity, the right to withdraw from research and anonymity are a matter of course. We will try to think about how to improve this situation.

Kind regards

Reviewer 2 Report

Comments and Suggestions for Authors

The Authors of the manuscript have mainly satisfied with the requests of my review.

Concerning the Introduction

A proper section was devoted to the elaboration of the theoretical and empirical introduction. They have explained the reasons for lack of peer-reviewed studies addressing the psychological variables of tetanic syndrome, which results in the exploratory nature of study, and have added the practical relevance of the findings to the Discussion section, emphasized the limitations.

They have also added research questions and some hypotheses.  

Concerning Psychometrics:

The Authors have replied sufficiently to my observations regarding the tools used.

They have included the limitations resulting from the use of the self-report method, online collection, and snowball sampling in the appropriate section.

Research Sample Description

They have added data regarding the collection period and criteria for inclusion in the research.

Author Response

Dear Reviewer,
We sincerely thank you for your positive opinion and for your positive evaluation of our efforts to revise the paper. We know that the first version had many shortcomings (though we found errors in the interpretation of the results in the discussion, which we corrected), and thanks to your comments, the article became more valuable and worthy of publication. We greatly appreciate this opportunity.

Best regards

Reviewer 3 Report

Comments and Suggestions for Authors

The recruitment process is still described insufficiently (which social media groups were examined?).

This study does not follow rigorous clinical methodology. In general, the diagnoses were not checked by professional practitioners. The information was collected in an online study, where people self-reported their diagnoses.

As it was indicated previously, including 2 people who did not from when they had tetany syndrome seems questionable. If they do not know from when they had this disease, how can we be sure that they actually have it.

Minor:

The form of the authors' reply is prepared not according to journal's requirements. Usually, the reviewer's comments are supplemented by the authors reply. However, in this case, the authors just posted their replies. Such a form makes assessment of the revisions more difficult. 
